# Effectiveness of Food Fortification in Improving Nutritional Status of Mothers and Children in Indonesia

**DOI:** 10.3390/ijerph18042133

**Published:** 2021-02-22

**Authors:** Nikmah Utami Dewi, Trias Mahmudiono

**Affiliations:** 1Department of Nutrition, Faculty of Public Health, University of Tadulako, Palu 94148, Indonesia; 2Department of Nutrition, Faculty of Public Health, University of Airlangga, Surabaya 60115, Indonesia

**Keywords:** fortification, Indonesia, nutritional status, stunting

## Abstract

Food fortification programs have been conducted in several countries to overcome micronutrient deficiency and related problems with various degrees of effectiveness. Available information regarding the success of food fortification programs in some developing countries, including Indonesia, is still limited. Thus, this study conducts a systematic review of the effects of food fortification of mothers and children using biochemical and anthropometric measures focusing on linear growth. Three databases were used in the literature search, namely PubMed, Science Direct and Google Scholar. Fifteen articles were included for analysis from 517 studies found consisting of Indonesian and English articles published from 2000 to June 2020. Fortification of iron, vitamin A, and iodine can increase the level of hemoglobin, serum ferritin, and serum retinol and median urine iodine excretion, especially in toddlers and schoolchildren. However, multinutrient fortification interventions were associated with various effects on hemoglobin, serum ferritin, and serum retinol but a positive association was found with linear growth indicators in the form of body length for age. The effectiveness of food fortification in reducing the prevalence of stunting still needs more and stronger evidence through studies with large sample size and longer duration.

## 1. Introduction

Hidden hunger is a form of micronutrient deficiency that is still experienced by various developing countries, including Indonesia [1]. In Indonesia, in 2018, the percentage of children with anemia was 38.5%, which increased by 10.4% compared with that in 2013 [2,3]. The percentage of individuals with anemia among pregnant women was much higher, 37.1% in 2013, and increased to 48.9% in 2018 [2,3]. The high prevalence of anemia in Indonesia is mostly related to low iron intake or infectious diseases that increase iron loss from the body, leading to iron deficiency [4,5,6,7]. Iron deficiency is a major cause of anemia [2,3] and a risk factor for zinc deficiency that can result in stunting [8,9].

One of the biggest nutritional problems in Indonesia is stunting. Although the prevalence of stunting among children under 5 years old in Indonesia has decreased from 37.2% to 30.8% [2,3], the number of cases still raises a public health concern [10]. The serious implications of stunting experienced by children and its impact on Indonesia’s development have made stunting reduction a national priority with a target of decreasing it to 19% by 2024 [11].

A program aiming to reduce community nutritional problems, including stunting, should be multisector and consistent with specific and sensitive programs [12]. Specific programs include short-term interventions for which results can be recorded in a relatively short time using activities performed by the health sector, whereas sensitive programs are long-term interventions in the form of activities that are mostly macro and performed across institutions [12]. Food fortification programs are an example of sensitive programs conducted in several countries [12]. Food fortification programs conducted worldwide included fortification of vitamin A in cooking oil, margarine, and sugar; vitamin D in milk and margarine; folic acid in flour; iodine in salt; iron in milk, corn flour, beans, pearl millet, and wheat flour [13,14,15,16]. In Indonesia, the most critical fortification interventions are iodine in salt and iron fortification of wheat flour, whereas the fortification of vitamin A in cooking oil was optional, but now compulsory since 2020 [17,18,19].

Food fortification in several countries is associated with different effects on nutritional status or the reduction of the prevalence of nutritional problems. Iron fortification in foodstuffs is associated with increased hemoglobin and serum ferritin levels and decreased the prevalence of anemia in children, pregnant women, adolescents, and adults [16,20,21]; however, it was not positively related to a decrease in stunting and malnutrition in children under 5 years old [22]. Fortification of vitamin A in oil, margarine, sugar, and processed foods increased serum retinol levels [23,24,25], whereas fortification of vitamin A in staple foods showed the opposite result [26]. Fortification of salt decreased the prevalence of goiter and cretinism and increased the median urinary excretion of iodine [27], whereas fortification of foods other than salt increased urinary excretion of iodine but not height growth based on child age [28].

We are focusing on the fortification of micronutrients, namely, vitamin A, iron, and iodine, in a single food or with other micronutrients. These micronutrients were the focus of fortification in Indonesia until 2020 to reduce the prevalence of vitamin A deficiency, anemia, disorders due to iodine deficiency (IDD), and stunting [29]. In Indonesia, several studies have been conducted to examine the correlation between food fortification and nutritional status [30,31,32,33,34,35,36,37,38,39,40,41]. However, a meta-analysis or systematic review has never been conducted. We systematically reviewed the literature from the past 20 years published in English and Bahasa Indonesia to determine the association between food fortification programs in Indonesia and improvement of nutritional status in mothers and children using biochemical and anthropometric measures related to stunting. 

## 2. Materials and Methods

This was an unregistered systematic review aimed to assess the effects of food fortification of mothers and children using biochemical and anthropometric measures focusing on stunting. Literature was searched using the PubMed and ScienceDirect databases for English-language studies and Google search engine for Indonesian-language studies. The keywords used were “fortification,” “Indonesia,” “nutrition,” “anemia,” “vitamin A,” “iron,” “iodine,” “malnourished,” and “stunting.” Articles used only in English and Indonesian. Using these keywords, we obtained 517 articles on PubMed search, 179 articles in ScienceDirect, and 339 articles in Google including Google Scholar. In addition, article searches were performed on the selected article bibliography to ensure that all relevant articles had been included in the review list. 

Articles were assessed by title and abstract based on certain inclusion and exclusion criteria. The identification of titles and abstracts was performed to see the suitability of the data presented with the stated objectives. The results of food fortification studies using nutritional status as the dependent variable were included in the review. The fortification included were vitamin A, iron, and iodine, into a single food or with other micronutrients. Nutritional status included anthropometry consisting of weight, height/body length, weight for age, height/body length for age, birth weight, knee height, and biochemical measures consisting of hemoglobin, serum retinol, serum ferritin, urine iodine excretion, and hematocrit. The studies were published from 2000 to June 2020 involving study samples consisting of groups of children, women and pregnant women with their babies. The research designs of the studies reviewed consisted of cross-sectional, cohort, quasi-experimental, randomized controlled trial (RCT), and nonrandomized controlled trial (Non-RCT) designs. Articles not in English nor Indonesian and those that did not study anemia, vitamin A deficiency, and iodine deficiency were excluded from this review. The articles were also excluded if they did not contain nutritional status measurements. Articles with conference abstracts, not original studies and containing information similar to the other articles were also excluded (Figure 1). Home fortification involving the use of Sprinkle was not included in this review because it was not conducted on the same type of food.

The articles that met the inclusion criteria were coded and are summarized in Table 1, including information on the study design, sample size, and participants, and Table 2, including duration, intervention, control provided, and the results obtained. In total, 15 articles were included according to the criteria used in this review.

## 3. Results

The 15 studies consisted of five RCTs [30,36,38,39,42], four quasi-experimental studies [32,33,37,43], four cohort studies [31,34,40,41], one cross-sectional study [44], and one non-RCT [35]. Most studies were conducted in the West Java region [31,34,38,39,40,41,42,44]. Three studies were conducted in Central Java [36,37,43], two studies in South Sulawesi [32,35], and one study in Jakarta City [30] and one used single samples from several cities in Indonesia [44] (Table 1).

Two studies assessed the effectiveness of iron fortification [30,31], three studies assessed the effectiveness of vitamin A fortification [32,33,34], three studies assessed iodine fortification [35,36,37], and eight studies assessed multinutrient fortification, including the effectiveness of iron fortification with the addition of other nutrients [31,38,39,40,41,42,43,44]. Fortified food ingredients included oil [32,33,34], candies [30], biscuits [38,39,40,41,43], vermicelli [39,40,41], instant noodles [44], milk [39,40,41,44], rice [42], salt [36,37], eggs [35], and baby food [31] (Table 2). Anthropometric research parameters were body weight [43], birth weight [39], body weight based on age [31], body weight based on body length [31,38], height or length based on body weight [31,41,44], and knee height. The biochemical nutritional status measurements included albumin level [43], serum retinol [32,33,34,38,39], hemoglobin [30,32,34,38,39,41], hematocrit [41], serum ferritin [30,39], and urine iodine excretion (UIE) [35,36,37]. The duration of the interventions or the length of the studies varied from 10 days to 1 year.

As seen in Table 2, iron fortification in candies at a dose of 30 mg per week for 3 weeks increases hemoglobin and serum ferritin levels in children aged 4–6 years old [30]. The assessment of vitamin A fortified cooking oil in two provinces found different results. Studies have reported a significant increase in serum retinol levels, especially in the groups of toddlers and children aged 6–59 months and 5–9 years, respectively, in West Java after fortification [30,40]. In a study conducted in Makassar City, no difference in serum retinol level was found between groups with and without the consumption of Vitamin A-fortified cooking oil [32]. Fortification of iodine in eggs for 10 days significantly increased the median UIE in children aged 10–12 years with iodine deficiency [35], whereas the fortification of iodine in salt for 6 months increased the median UIE but not significantly in children aged 6–12 years in Central Java [36]. A quasi-experimental study in Central Java showed a decrease in UIE after 6 months of fortification of iodine in salt in children aged 4–9 years [37].

Multinutrient fortification interventions were associated with varied results on nutritional status. Vitamin A and iron-fortified biscuit consumption increased the hemoglobin and serum retinol levels in children under 5 years old in a study conducted in Bogor Regency [38]; however, a study conducted in Semarang showed that iron and zinc fortification in tempeh rice-bran biscuits were associated with increased albumin levels and body weight, that were not significantly different from those of the control group [43]. A study conducted in Bogor District on pregnant women showed that the fortification of multinutrients in biscuits, vermicelli, and milk increased the hemoglobin levels but not the mothers’ serum ferritin levels [39]. In addition, there was no significant association found between the hemoglobin levels and fortified food in the babies born to mothers with and without fortified food [41].

Regarding linear growth, a study reported that iron fortification in foods in a 9-month-old baby for 84 days (3 months) were associated with increased height for age at 12 months [31]. Among 6-month-old infants, the body lengths and body length gains were significantly higher in the group whose mothers were given multiple-nutrients-fortified foods compared to those in the control group [40]. Studies on children under 5 years old consuming fortified foods showed that foods fortified with multinutrients can increase the body length for age to prevent stunting [31,44] (Table 3).

## 4. Discussion

This review involved 15 studies conducted in the past 20 years. Most studies were conducted in Java Island, especially in the West Java region. West Java Province is the region with the largest population in Indonesia [45]. The longest study duration was 1 year. The average sample size of the studies reviewed was 100–200 individuals, with the largest study sample of 302,190 individuals in one study [44]; therefore, the large number of samples in this study could affect the power of the study [45]. Most studies assessed the effectiveness of multinutrient fortification using dairy as the most widely used food vehicle. The most widely performed biochemical parameters were hemoglobin and serum retinol, whereas the main anthropometric measure was height or body length for age. 

Our review revealed that food fortification is effective in increasing hemoglobin and serum ferritin levels and decreasing the prevalence of anemia in infants, children, and toddlers [20,46]. In pregnant women, iron fortification could reduce the risk of preterm birth and giving birth to babies with low birth weight [21]. An RCT in Jakarta revealed that iron fortification in foodstuffs can increase the hemoglobin and serum ferritin levels of children aged 4–6 years with a weekly dose of 30 mg of iron for 3 weeks [30]. However, studies in West Java showed that fortified rice with an iron dose of >50 mg/kg could not increase the hemoglobin and serum ferritin levels in schoolchildren aged 12–15 years [42]. The results of RCTs are often different from those of effectiveness studies. Thus, these results need to be interpreted carefully, especially in evaluating the success of iron fortification of food, which is practiced in large scale. In Indonesia, iron fortification of wheat flour became mandatory starting in 2001, whereas iron-fortified rice began to be promoted by BULOG in 2019 [18]. The required dosage of flour fortification according to the Presidential Regulation of the Republic of Indonesia is at least 50 mg/kg of iron along with a minimum of 30 mg/kg of zinc, 2.5 mg/kg of thiamine, 4 mg/kg of riboflavin, and 2 mg/kg of folic acid [47]. 

Regarding anthropometric outputs, iron deficiency is closely related to the incidence of stunting. Iron functions in skeletal growth through the formation of collagen and metabolism of vitamin D which is required in bone formulations [48,49]. Iron deficiency impacts bone homeostasis through disruption of osteoclast and osteoblast activity and differentiation [50]. One study showed that iron fortification of infant foods could increase the body length based on the age of 12-month-old infants [31]. Studies conducted in other countries have revealed that iron fortification may improve iron status in the body but not body length for age [51,52,53] and may increase height for age but not significantly [54]. The effect of fortification was greater in subjects with anemia at baseline than that in those with normal condition [55]. 

Another form of food fortification found in Indonesia based on the results of our review was a nutrient improvement in cooking oil through vitamin A fortification. The effectiveness of vitamin A fortification in improving vitamin A status in Indonesia was evaluated in two studies. An evaluation study of the use of fortified cooking oil in Makassar City, South Sulawesi, for 3 months did not show changes in serum retinol in the intervention group [32]. However, this study showed that the prevalence (26.6%) of vitamin A deficiency was lower in children who consumed fortified oil for ≥12 weeks compared to children who consumed fortified oil for less than 12 weeks (42%) [32]. A study involving infants, girls, and breastfeeding mothers in West Java for 1 year at a dose of 13.6 mg retinol/kg of food vehicle showed that vitamin A-fortified cooking oil significantly increased serum retinol [33]. The longer duration of fortified oil consumption affects the storage of vitamin A in the body, thus increasing vitamin A status [56]. Positive results have been also found in several countries regarding the effects of fortification of food with vitamin A in increasing serum retinol levels [26].

Another nutrient that was widely used in food fortification was Iodine. Iodine fortification in Indonesia began during the Dutch occupation era in 1927, stopped in 1945, and restarted in 1976 [57]. Moreover, the Urine Iodine Excretion (UIE) of schoolchildren increased from 164.8 µg/L in 1995 to 330.2 µg/L in 1997 and 306.0 µg/L in 1999 [57]. The schoolchildren in only one province showed UIE of <100 µg/L in 1999 [57]. A quasi-experimental study in Central Java in 2015 was not associated with an increase in median UIE or a decrease in the prevalence of iodine deficiency of <100 µg/L UIE in children aged 4–9 years [37]. RCTs in different parts of central Java using fortified iodine doses of 15–55 mg/kg increased the UIE in children 6–12 years, but this did not reach statistical significance [36]. One study in South Sulawesi showed that the fortification of eggs with iodine for 10 days at a fortification dose of 0.4 mg/kg significantly increased the median UIE in children aged 10–12 years [35]. Quasi-experimental studies and RCTs conducted in Central Java found that the consumption habits of iodine sources were significantly different between groups [36,37], thus potentially affecting the results obtained.

The multi-micronutrient interventions in this review showed various results in improving the biochemical nutritional status in mothers and children. Vitamin A- and iron-fortified biscuits could increase the hemoglobin and serum retinol levels in toddlers [38], whereas the fortification of biscuits, vermicelli, and milk with multinutrients could increase the hemoglobin but not serum ferritin levels of mothers and showed no effect on the hemoglobin levels of babies [39,40]. In pregnant women, the phenomenon of decreasing serum ferritin may occur due to the use of ferritin in increasing the mass of maternal blood cells [58]. In studies conducted in various countries, the provision of multinutrient-fortified foods has a positive effect on reducing the prevalence of anemia and iron deficiency in mothers [59]. Two studies indicated that multinutrient fortification interventions had a significant positive influence on the body length growth and the z score for body length based on infant age [31,40]. In addition, a study showed that multinutrient fortification interventions reduced the prevalence of stunting in toddlers [44]. The effectiveness of multinutrient fortification on improving the linear growth in Indonesia can be explained by the pre-existing multinutrient deficiencies that may be experienced by individuals in the population [60]. Since macronutrient deficiencies such as wasting and underweight remain a major public health problem in Indonesia [2], protein and energy malnutrition likely affect the body’s physiological functions [61]. Lack of these nutrients causes rapid growth failure because micronutrients cannot be of maximum benefit according to their function in linear growth [61,62]. 

Multinutrient fortification had a positive impact on the linear growth although the impact is still insignificant in children [63,64]. Nevertheless, the statistically insignificant result does not mean that the change was not biologically significant to the health of the individuals, particularly with respect to children. The difference in results from those found in Indonesia could be due to differences in the duration of the intervention performed [54,55], the type of vehicle used [51,54], the target group in the study [54], or pre-existing micronutrient deficiency conditions that were unknown at the start of the study [55].

Although food fortification in Indonesia showed promising results in improving nutritional status and decreasing the prevalence of anemia, iron deficiency, and vitamin A deficiency, various studies with longer durations of body height escalation assessment involving large study samples are needed to discover the effectiveness of food fortification on body height escalation and the reduction of stunting prevalence. In addition, this study did not evaluate the types of fortification methods used due to the lack of information available from the studies reviewed. To the best of our knowledge, this study is the first systematic review describing the effectiveness of nutrient fortification in food in improving the nutritional status in children and pregnant women in Indonesia.

## 5. Conclusions

The effectiveness of food fortification in reducing micronutrient deficiency problems in Indonesia presents promising results; however, the effectiveness of food fortification in reducing the prevalence of stunting still needs more and stronger evidence, although several studies have indicated positive results. Nonetheless, this review might be a starting point for a sound strategy for future studies pertaining to the level and duration of food fortificant used.

## Figures and Tables

**Figure 1 ijerph-18-02133-f001:**
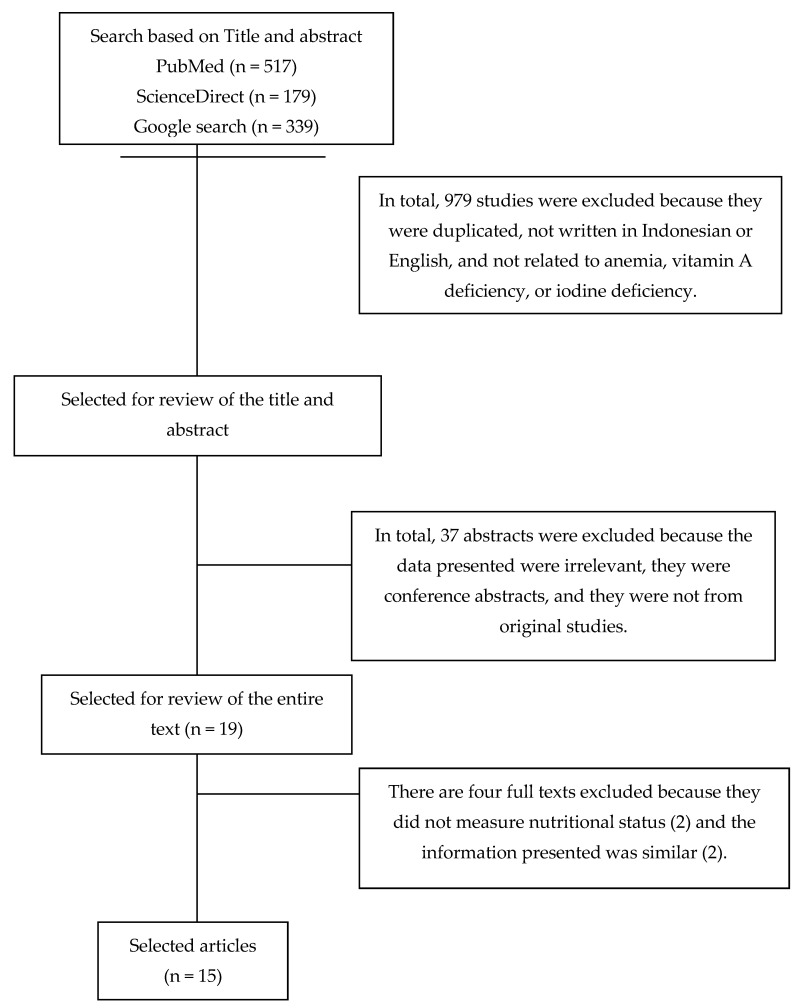
CONSORT diagram of the literature search and determination.

**Table 1 ijerph-18-02133-t001:** Studies on the relationship between food fortification and nutritional status.

No.	Reference	Year	Study Design	Sample Amount (*n*)	Group	Location (Province)	Duration (Days)
Iron							
1.	Sari [30]	*2001*	*RCT*	132	4–6 years old	Jakarta	21
2.	Diana [31]	*2017*	*Cohort*	190	6 months followed up to 12 months old	West Java	84
Vitamin A							
1.	Achadi [32]	*2010*	*Quasi-experimental*	394	8–9 years old	South Sulawesi	84
2.	Sandjaja [33]	*2015*	*Quasi experimental*	Breastfeeding mother: 335, 12–59 months old: 477, 5–9 years old: 186, woman 15–29 years old: 171	Breastfeeding mother, baby aged 6–11 months, children aged 12–59 months, children aged 5–9 years, women aged 15–29 years	West Java	336
3.	Sudikno [34]	2017	Cohort	126	Children of poor family, 6–59 months old	West Java	336
Iodine							
1.	Kasmawati [35]	*2015*	*Non-RCT*	26	10–12 years old with urinary iodine excretion value of <100 μg/L	South Sulawesi	10
2.	Samsudin [37]	*2015*	*Quasi-experimental*	160	4–9 years old	Central Java	168
3.	Samsudin [36]	2016	RCT	176	6–12 years old	Central Java	168
Multinutrient							
1.	Widayanti [38]	*2007*	*RCT*	70	≤5 years old	West Java	112
2.	Prihananto [39]	*2007*	*RCT*	210	Pregnant mothers	West Java	168
3.	Saragih [40]	*2007*	*Cohort*	120	Pregnant mothers with babies up to 6 months old	West Java	336
4.	Saragih [41]	*2012*	*Cohort*	120	Pregnant mothers with babies up to 6 months old	West Java	336
5.	Kurnia [43]	*2010*	*Quasi-experimental*	91	≤5 years old	Central Java	84
6.	Semba [44]	*2011*	*Cross-sectional*	302,190	6–59 months old	West Sumatera, Lampung, Banten, Jakarta, West Java, Central Java, East Java, West Nusa Tenggara, South Sulawesi	-
7.	Diana [31]	*2017*	*Cohort*	190	6 months followed up to 12 months old	West Java	84
8.	Toruntju [42]	2017	RCT	80	Boys, 12–15 years old, with Hb concentration of 8–12 mg%	West Java	168

**Table 2 ijerph-18-02133-t002:** Type and level of fortification and food vehicle.

Study	Type and Level of Fortification	Food Vehicle
Iron		
Sari [30]	Iron 30 mg, weekly dose	Candies
Diana [31]	Iron	Baby food
Vitamin A		
Achadi [32]	Vitamin A: 13.6 mg retinol/kg	Cooking oil
Sandjaja [33]	Vitamin A: 13.6 mg retinol/kg	Cooking oil
Sudikno [34]	Vitamin A: 13.6 mg retinol/kg	Cooking oil
Iodine		
Kasmawati [35]	Iodine: 0.4 mg/kg KIO3	Eggs
Samsudin [37]	Iodine: ≥30 mg/kg KIO3	Salt
Samsudin [36]	Iodine: 15–55 mg/kg KIO3	Salt
Multinutrient		
Widayanti [38]	235.65 μg of vitamin A and 4.17 mg of iron/100 g of biscuits. 76 g in a week	Biscuits
Prihananto [39], Saragih [40], Saragih [41]	Biscuit/100 g: iron: 16 mg; iodine: 36.76 mcg; zinc: 6.3 mg; folic acid: 66.72 mcg; vitamin A: 345.76 RE; vitamin C: 46.39 mg Vermicelli/100 g: iron: 4.4 mg; iodine: 18.27 mcg; zinc: 4.4 mg; folic acid: 159.56 mcg; vitamin A: 494.906 RE; vitamin C: 45,27 mg Milk/100 g: iron: 22.58 mg; iodine: 58.40 mcg; zinc: 3.29 mg; folic acid: 48.55 mcg; vitamin A: 468.19 RE; vitamin C: 127.2 mg	Vermicelli, milk, biscuits
Kurnia [43]	Iron and zinc 10 mg/100 g biscuits. 300 g in a week	Tempeh rice-bran biscuits
Semba [44]	Vitamin A, vitamin C, vitamin D, vitamin E, vitamin K, vitamin B12, thiamine, and riboflavin Vitamin B6, niacin, folic acid, and iron.	Milk and instant noodles
Diana [31]	Iron, zinc, calcium, vitamin A	Baby food
Toruntju [42]	Iron, Zn, vitamin B1, vitamin B3, folic acid, and vitamin B12	Rice

**Table 3 ijerph-18-02133-t003:** Fortification outcome in several studies.

Study	Outcome			
	**Nutritional Status**		**Prevalence (%)**	
	**Nonfortification Group**	**Fortification Group**	**Nonfortification Group**	**Fortification Group**
Iron				
Sari [30]	∆Hb (g/L)		Anemia (Hb < 11 g/dL)	
	↑4.0 (95%CI: 2.0–6.0)	↑10.2 (95%CI: 8.3–12)	↓16.6	↓48.9 *
	SF (μg/l)		ID (SF < 12 μg/L)	
	↑28%	↑71%	↓35.1	↓13.4
Diana [31]	BL/A		n/a	
		Iron: β = 0.22; 95% CI: 0.01–0.44 Others: β = 0.29; 95%CI: 0.09–0.48		
	W/BL (kg/cm)			
		Iron: β = −0.22; 95%CI: −0.42–0.00 Others: β = −0.09; 95%CI: −0.29–0.11		
	W/A			
		Iron: β= −0.03; 95%CI: −0.15–0.10 Others: β = 0.14; 95%CI: 0.02–0.26		
Vitamin A				
Achadi [32]	SR (μg/dL)		Anemia (Hb < 11.5 g/dL)	
	23	22.2	21.8	11.6.
			VAD (SR < 20 ug/dL)	
			38.4	38.8 ≥12 weeks: 26.6% <12 weeks: 42%
Sandjaja [33]	∆SR (μg/dL)		KVA (SR < 20 ug/dL)	
		6–11 months: 12.3 ** 12–23 months: 2.2 24–59 months: 6.4 5–9 years: 14.9 ** Breastfeeding mother: 13.1 ** Nonbreastfeeding mother aged 15–29 years: 19.1 **	6.5–18	0.6–6 *
Sudikno [34]	∆SR (μg/dL)		VAD (SR < 20 ug/dL)	
	29.36 ± 1.07	35.19 ± 0.89 **	19.0	5.6 *
	∆Hb (g/L)		Anemia (Hb < 11 g/dL)	
	11.18 ± 0.12	11.59 ± 0.14	43.7	28.6 **
Iodine				
Kasmawati [35]	∆UIE (μg/L)		n/a	
	6.38	12.4 **		
Samsudin [37]	UIE (μg/L)		ID (UIE < 100μg/L)	
	*Coastal: 191* *Noncoastal: 96*	*Coastal: 148** *Noncoastal: 83*	*Coastal: 10.4* *Noncoastal: 51.8*	*Coastal: 28.6* *Noncoastal: 60.2*
Samsudin [36]	UIE (μg/L)		ID (UIE < 100μg/L)	
	222	238	14	13
	TSH (μIU/mL)			
	1.9(0.5–4.0)	1.8 (0.2–5.9)		
Multinutrient				
Widayanti [38]	∆Hb (g/L)		Anemia (Hb < 11 g/dL)	
	↑0.17 ± 1.05	↑ 0.67 ± 1.11 **	↓ 20	↓ 22.8
	∆SF (μg/L)		IDA (SF < 12μg/L)	
	↑8.27 ± 4.55	↑ 11.43 ± 4.47 **	↓ 2.9	↓ 17.2
	∆SR (μg/dL)		VAD (SR < 20 ug/dL)	
	↑4.38 ± 7.72	↑ 10.12 ± 7.84 **	↓ 22.6	↓ 44.1
	∆W/L (kg/cm)		Wasting	
	−0.07 ± 0.66	−0.01 ± 1.07	14.3	5.7
Prihananto [39]	∆Hb (g/L)		Anemia (Hb < 11 g/dL)	
	Pl: ↓1, C: ↓1.1	↓ 0.2 *	Pl: 77.6, C: 86.2	48.3 *
	∆SF (μg/L)		∆IDA (SF < 12μg/L)	
	Pl: ↓0.25, C: ↓0.32	0.23 ^a^	Pl: ↑ 71.7, C: ↑ 63.3	↑ 39.4
	BW (kg)		VAD (SR < 20 ug/dL)	
	Pl: 3.06 ± 0.34, C: 2.98 ± 0.32	3.01 ± 0.27	Pl: 20.3, C: 39.7	17.2
			LBW (BW < 2.5 kg)	
			Pl: 5.3, C: 3.6	0
Saragih [40]	∆BL (cm)		n/a	
	Pl: 16.41 ± 1.41 C: 15.76 + 1.70	17.94 ± 1.83 *		
	∆BL/A			
	Pl: −0.26 ± 0.88 C: −0.52 ± 1.16	0.50 ± 0.92 *		
	∆KH (cm)			
	Pl: 4.02 ± 0.71 C: 3.80 ± 0.74	4.47 ± 0.64		
Saragih [41]	Hb (g/L)		Anemia (Hb < 11 g/dL)	
	Pl: 83.0–123.2 C: 70.6–120.3	95.7–120.0	Pl: 44.4 C: 63.9	47.2
			Anemia (Ht < 33%)	
			Pl: 30.6 C: 38.9	27.8
Kurnia [43]	Albumin (g/dL)		n/a	
	Pl: 1.09 ± 1.38C: 0.92 ± 0.41	0.95 ± 0.50		
	W (kg)			
	Pl: 0.99 ± 1.45C: 0.24 ± 0.65	0.61 ± 2.34		
Semba [44]	Milk			
		Stunting Village, OR = 0.87; 95% CI, 0.85 to 0.90 City, OR = 0.80; 95% CI, 0.76 to 0.85	Village 56.2 City 53.7	Village 43.4 ** City 42.8 **
	Noodles			
		Village, OR = 0.95; 95% CI, 0.91 to 0.99 Jakarta, OR = 0.95; 95% CI, 0.91 to 1.01	Village 53.6 City 51.5	Village 45.6 ** City 45.9 **
Toruntju [42]	∆Hb (g/L)		n/a	
	↑0.7	↑ 0.41		
	∆SF (μg/L)			
	↑0.77	↓ 9.94		
Sudikno [34]	∆SR (μg/dL)		VAD (SR < 20 ug/dL)	
	29.36 ± 1.07	35.19 ± 0.89 **	19.0	5.6 *
	∆Hb (g/L)		Anemia (Hb < 11 g/dL)	
	11.18 ± 0.12	11.59 ± 0.14	43.7	28.6 **

* *p* < 0.05 between the fortification and nonfortification group, ** *p* < 0.001 between fortification and nonfortification group. ^a^
*p* < 0.05 between fortification and placebo. IDA, iron deficiency anemia; ID, iodine deficiency; BW, birth weight; BL, body length; LBW, low birth weight; BL/A, body length for age; W/BL, weight for body length; W/A, weight for age; HB, hemoglobin; SR, serum retinol; SF, serum ferritin; UIE, urine iodine excretion; Ht, hematocrit; KH, knee height; Pl, placebo group; C, control group.

## Data Availability

Availability of the data will be release upon request through corresponding authors.

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
