# Peer review of "Effectiveness of Food Fortification in Improving Nutritional Status of Mothers and Children in Indonesia"

_ijerph, 2021, doi:10.3390/ijerph18042133_

Round 1

Reviewer 1 Report

Overall, this review has improved tremendously from my first review. The topic is very important and will greatly contribute to the literature on the validity of fortification programs in Indonesia on women and children. Only a few comments below.

Line 47-50 –Again, if you are thoroughly reviewing the literature, you need to include iron fortified beans and pearl millet. Here is a reference for adult women to include:

Haas, Jere D., et al. "Consuming iron biofortified beans increases iron status in Rwandan women after 128 days in a randomized controlled feeding trial." The Journal of nutrition 146.8 (2016): 1586-1592.

Your reference #20 has information in pearl millet biofortification and an RCT in children.

Line 66-67 – provide references

Line 160 – restate you general purpose “We systematically reviewed the literature to determine 69 the association between food fortification programs in Indonesia and improvement of nutritional 70 status in mothers and children using biochemical and anthropometric measures related to stunting” and tie it in with the 15 reviews conducted over 20 years. Be more thorough, as written it is too generic.

Line 248-252 – make sure to state “in Indonesia” in this sentence… this is what you evaluated. This also should be included in your conclusion paragraph.

Noticed a few spelling/grammar errors.

Reviewer 2 Report

The paper improved very much now, but to clarify my first review:

-keywords were meant if they were used for the literature search as single keywords or also in combinations. Has nothing to do with nutrients.

It would be helpful to use consistent parameters, e.g. for iron and others there ist mg and ppm used. Stay consistent.

Line 197: not vitamin A was fortified but a nutrient with vitamin A!

Language still has to be improved, e.g.:

line 141: "no significant association were found"

line 234: "Lack of macro nutrients such as wasting or underweight also still a public health problem in Indonesia [2]."

Author Response

This manuscript is a resubmission of an earlier submission. The following is a list of the peer review reports and author responses from that submission.

Round 1

Reviewer 1 Report

Overall, this review has potential. The topic is very important and would greatly contribute to the literature on the validity of fortification programs in Indonesia on children. Unfortunately, this manuscript needs more focus, clarification, and quite a few revisions to strengthen it. In general, each section needs some work. For the methods, the search process needs to be more detailed and justification for your inclusion/exclusion process needs to be detailed. It also is unclear on how many manuscripts were part of this review or whether the review is on children, WRA, or both. The discussion needs significant revisions as well; based on the abstract, this review is focused in children. The discussion mentions many studies that include women, but then it is unclear if the next sentences are referring to women or kids. Overall, the discussion rambles and needs to be rewritten to focus on the chosen population and the fortification of vitamins in food on the stated outcomes.

Again, this review has a ton of potential! Please revise and resubmit.

Abstract – rewrite last sentence, awkward

Line 47-49 –awkward wording

Line 51-52 – iron has also been fortified in beans; review and reference these authors à Jere D. Haas/Laura Murray-Kolb/Saurabh Mehta/Julia Finkelstein to name a few; most of these studies are RCTs in adults, so if you are referring to fortification in children, then please be more specific.

Line 65-72 – Again, are you referring to children only? This is very unclear the way it is written. This entire paragraph could use some help; it is poorly written.

Line 67 – provide reference, also awkward sentence; small sample size

Line 70 and 74 – use the word “review” instead of “study”

Line 75 – use “included in the review” somewhere; this sentence is poorly written as is.

Line 76-78 – this sentence should be at the end of the Intro, not in the Methods

Paragraph (lines 74-84) –rewrite this entire paragraph. Right now it reads as you are including more than just kids. This entire paragraph should include what micronutrients you focused on (any specific amounts in the foods or specific foods), what outcomes were you looking at such as Hb, sFer, Vit A, height, weight, etc. (this should be more specific than anthropometry and biochemistry), what articles were excluded and why, etc. Be specific on what was included and excluded here beyond just the types of studies that you included in the review.

Line 86&89 – assuming you mean Google Scholar?

Paragraph (lines 91-100) – be more specific on your selection process

Line 97-99 – this sentence does not make sense as written

Line 99, lines 104&105 – is it 10 or 15? Lines 104-5 indicate 14?

Line 121&122 – use “30 mg per week”; also, “could” is the wrong word

Author Response

Response to Reviewer 1 Comments

Abstract – rewrite last sentence, awkward

Response: Thank you for your feedback. The last sentence has been rewritten. We would like to convey that the results we get from the review need to be confirmed by studies with a larger sample size and sufficient time

Line 47-49 –awkward wording

Response: We want to explain further about a fortification which is a program aimed at overcoming malnutrition. However, we understand that this definition is generally known. So we delete this sentence and move the paragraph up to be part of the third paragraph in the introduction as they have the same idea.

Line 51-52 – iron has also been fortified in beans; review and reference these authors à Jere D. Haas/Laura Murray-Kolb/Saurabh Mehta/Julia Finkelstein to name a few; most of these studies are RCTs in adults, so if you are referring to fortification in children, then please be more specific.

Response: Thank you for your suggestion, in line 51-52, we described programs that have been carried out in several countries related to food fortification. The programs and activities were aimed at particular country. We add the reference of Haas et al to be included in the following paragraphs on line 56-59 which discuss the effectiveness of fortification that has been carried out in various groups.

Line 65-72 – Again, are you referring to children only? This is very unclear the way it is written. This entire paragraph could use some help; it is poorly written.

Response: This review focuses on children, but the research we involved also involved pregnant women and their children so that the outcome of mother child's birth could be seen (e.g birth weight). The final paragraph of introduction section has been reformulated to provide general overview of the purpose of the review.

Line 67 – provide reference, also awkward sentence; small sample size

Response: Thank you for your suggestion. Reference was provided and sentence was revised.

Line 70 and 74 – use the word “review” instead of “study”

Response: Thank you for your suggestion. We changed the word “review” instead of “study” as suggested.

Line 75 – use “included in the review” somewhere; this sentence is poorly written as is.

Response: Thank you for your suggestion. We revised the sentence as suggested.

Line 76-78 – this sentence should be at the end of the Intro, not in the Methods

Response: Thank you for your suggestion. We moved the sentence at the end of the introduction as suggested.

Paragraph (lines 74-84) –rewrite this entire paragraph. Right now it reads as you are including more than just kids. This entire paragraph should include what micronutrients you focused on (any specific amounts in the foods or specific foods), what outcomes were you looking at such as Hb, sFer, Vit A, height, weight, etc. (this should be more specific than anthropometry and biochemistry), what articles were excluded and why, etc. Be specific on what was included and excluded here beyond just the types of studies that you included in the review.

Response: Thank you for your suggestion. We rewrite the entire paragraph to be specific only towards children outcome.

Line 86&89 – assuming you mean Google Scholar?

Response: Not only google scholar but google search engine because some reports in language may not appear in google scholar.

Paragraph (lines 91-100) – be more specific on your selection process

Response: Thank you for your suggestion, we have provided the information about inclusion and exclusion criteria in the method section.

Line 97-99 – this sentence does not make sense as written

Response: Thank you for your input, we rewrite this section (method) to provide clear information from the search strategy to study selection criteria.

Line 99, lines 104&105 – is it 10 or 15? Lines 104-5 indicate 14?

Response: Thank you for catching this. It actually 15, We miss one study that should be put under the quasi-experimental study, we had referred to it in the text.

Line 121&122 – use “30 mg per week”; also, “could” is the wrong word

Response: Thank you for your suggestion. We revised the sentence as suggested.

Overall, this review has potential. The topic is very important and would greatly contribute to the literature on the validity of fortification programs in Indonesia on children. Unfortunately, this manuscript needs more focus, clarification, and quite a few revisions to strengthen it. In general, each section needs some work. For the methods, the search process needs to be more detailed and justification for your inclusion/exclusion process needs to be detailed. It also is unclear on how many manuscripts were part of this review or whether the review is on children, WRA, or both. The discussion needs significant revisions as well; based on the abstract, this review is focused in children. The discussion mentions many studies that include women, but then it is unclear if the next sentences are referring to women or kids. Overall, the discussion rambles and needs to be rewritten to focus on the chosen population and the fortification of vitamins in food on the stated outcomes.

Response: We make clear in the paragraphs that the review is only involving mother and children

Reviewer 2 Report

The article about **Effectiveness of Food Fortification in Improving Nutritional Status in Indonesia** should have some scientific meanings to the readers.

However, I have some concern about this review paper.

First, I suggest that the author should discuss the relationship about the nutrient deficiency or calories deficiency such as hidden hunger with stunting

Second, I suggest that the studies could be divided or grouped by the micronutrition- such as iron group, vitamin A group etc. Then we can understand the main idea of the article.   

Author Response

Reviewer 2 Comments

The article about **Effectiveness of Food Fortification in Improving Nutritional Status in Indonesia** should have some scientific meanings to the readers.

Response: Thank you for your assessment, we hope that the readership of IJERPH will benefit from this review articles, especially those working in developing countries.

However, I have some concern about this review paper.

First, I suggest that the author should discuss the relationship about the nutrient deficiency or calories deficiency such as hidden hunger with stunting

Response: Thank you for your suggestion, we add in the discussion section regarding the relationship between hidden hunger and macro and micronutrient deficiency in line 268-270 and line 282-287.

Second, I suggest that the studies could be divided or grouped by the micronutrition- such as iron group, vitamin A group etc. Then we can understand the main idea of the article.   

Response: Thank you for your suggestion, we rearange the studies based on the fortificant they used.

Reviewer 3 Report

The paper gives an important overview of the actual situation of studies dealing with food fortification in Indonesia which shows, as everywhere else, the general problem of nutritional epidemiological mstudies. There are always supportive as well as contradictory results, due to this special kind of studies with the huge number of variables. In addition, the relatively small distribution of the studyresults in international journals should be avoided. (Problem may be the rejecting rate of international journals for not so perfect studies!)

Anyhow, the paper gives a good overview which could serve as a starting point for the planning of future studies which for sure will take placer as the situation in Indonesia still is critical.

Minor points:

Some more info should be given for the use of keywords: have they only been used as singles or also incombinations?

Line 90: what about studies not mentioning "stunting"?

Line 99: here a tiotal of 10 studies is mentioned, must be a typing erreor as everywhere else 15 is indicated (e.g. line 101)

Lines 104-105: The listed types of studies sum up to only 14.

Line 117: are there informations if serum retinol only was measured or also RBP, as retinol is regulated homeostatically. (Just a question, no critics to the authors)

Line 127: this must be reference 34 (cf. lines 194 and201)

Ref. 34: are there infos on the concentration of Vit A in the supplement (table 2)?

In table 3 there seems to be enough space to indicate g/L or g/dl in the table and not in the table's footnotes

The discussion is well written and also conclusions in the diescussion are OK. However, in lines 192 to 201: if not looking at the references you might think to read about 3 studies, but it starts with 34, then comes 28 and the again 34. Please rearrange.

Line 207: it might be indicated that both groups in addition were from different localizations.

Line 212:  The data on UIE are not very helpful, as the endpoint 12,4µg/L is also below<100 as the value (not indicated) at baseline.

The conclusions are conclusive as mentuioned above, thie review might be a starting point for a good pülanning of future studies as regards e.g. readouts, concentrations of supplements and durations.

Author Response

Reviewer 3 Comments

The paper gives an important overview of the actual situation of studies dealing with food fortification in Indonesia which shows, as everywhere else, the general problem of nutritional epidemiological mstudies. There are always supportive as well as contradictory results, due to this special kind of studies with the huge number of variables. In addition, the relatively small distribution of the studyresults in international journals should be avoided. (Problem may be the rejecting rate of international journals for not so perfect studies!)

Anyhow, the paper gives a good overview which could serve as a starting point for the planning of future studies which for sure will take placer as the situation in Indonesia still is critical.

 Response: Thank you for your assessment, we hope that the readership of IJERPH will benefit from this review articles, especially those working in developing countries.

Minor points:

Some more info should be given for the use of keywords: have they only been used as singles or also incombinations?

Response: We are not sure about singles or combination that the reviewer reffer too. If the the reviewer means single or multinutrient fortification, we have include them in the table and text.

Line 90: what about studies not mentioning "stunting"?

Response: Thank you for your question. Study not mentioning stunting still included in the study unless they do not have outcome of nutritional status both antropometry and biochemical.

Line 99: here a tiotal of 10 studies is mentioned, must be a typing erreor as everywhere else 15 is indicated (e.g. line 101)

Response: Thank you for catching this. Yes, it was our mistake of typo. It should be fiveteen.

Lines 104-105: The listed types of studies sum up to only 14.

Response: It should be fiveteen. We have rewrite it as fiveteen. We have also added article that we refer to in line 104-105

Line 117: are there informations if serum retinol only was measured or also RBP, as retinol is regulated homeostatically. (Just a question, no critics to the authors)

Response: Thank you for your question. The information provided from the article was only Serum retinol. No information about RBP was available.

Line 127: this must be reference 34 (cf. lines 194 and201)

Response: Yes, you are right. Thank you for catching this. It have been revised accordingly

Ref. 34: are there infos on the concentration of Vit A in the supplement (table 2)?

In table 3 there seems to be enough space to indicate g/L or g/dl in the table and not in the table's footnotes

Response: No information directly stated about the consentration of vitamin A used in the article. But looking more detail to the reference and the history of oil they used from previous efficacy study, we found the concentration of vitamin A fortification they used.

The discussion is well written and also conclusions in the diescussion are OK. However, in lines 192 to 201: if not looking at the references you might think to read about 3 studies, but it starts with 34, then comes 28 and the again 34. Please rearrange.

 Response:  Thank you for your suggestion. We agree and sentence in line 192 to 201 have been rearange.

Line 207: it might be indicated that both groups in addition were from different localizations.

 Response: We have also also put additional information that the study in line 207 were from different location with previous study that stated in the text.

Line 212:  The data on UIE are not very helpful, as the endpoint 12,4µg/L is also below<100 as the value (not indicated) at baseline.

Response:  Thank you for your input. The right sentence should be “ with the increasing iodine levels of 12.4 µg/L”

The conclusions are conclusive as mentuioned above, thie review might be a starting point for a good pülanning of future studies as regards e.g. readouts, concentrations of supplements and durations.

Response:  Thank you for your input. The sentence “this review might be a starting point for a good planning of future studies pertaining concentrations and durations of food fortification and supplementation” heve been added to the conclussion section
